# A Targeted Proteomics Approach for Screening Serum Biomarkers Observed in the Early Stage of Type I Endometrial Cancer

**DOI:** 10.3390/biomedicines10081857

**Published:** 2022-08-02

**Authors:** Blendi Ura, Valeria Capaci, Michelangelo Aloisio, Giovanni Di Lorenzo, Federico Romano, Giuseppe Ricci, Lorenzo Monasta

**Affiliations:** 1Institute for Maternal and Child Health—IRCCS Burlo Garofolo, 34137 Trieste, Italy; valeria.capaci@burlo.trieste.it (V.C.); michelangelo.aloisio@burlo.trieste.it (M.A.); giovanni.dilorenzo@burlo.trieste.it (G.D.L.); federico.romano@burlo.trieste.it (F.R.); giuseppe.ricci@burlo.trieste.it (G.R.); lorenzo.monasta@burlo.trieste.it (L.M.); 2Department of Medicine, Surgery and Health Sciences, University of Trieste, 34129 Trieste, Italy

**Keywords:** biomarkers, proteins, endometrial cancer, targeted proteomics, PEA

## Abstract

Endometrial cancer (EC) is the most common gynecologic malignancy, and it arises in the inner part of the uterus. Identification of serum biomarkers is essential for diagnosing the disease at an early stage. In this study, we selected 44 healthy controls and 44 type I EC at tumor stage 1, and we used the Immuno-oncology panel and the Target 96 Oncology III panel to simultaneously detect the levels of 92 cancer-related proteins in serum, using a proximity extension assay. By applying this methodology, we identified 20 proteins, associated with the outcome at binary logistic regression, with a *p*-value below 0.01 for the first panel and 24 proteins with a *p*-value below 0.02 for the second one. The final multivariate logistic regression model, combining proteins from the two panels, generated a model with a sensitivity of 97.67% and a specificity of 83.72%. These results support the use of the proposed algorithm after a validation phase.

## 1. Introduction

Endometrial cancer (EC) arises in the inner part of the uterus and represents the fourth most common female malignancy in Europe [1]. Unlike other cancers, the incidence and mortality of EC are rapidly increasing worldwide, especially in North America and Western Europe (incidence 12.9–20.2 per 100,000 women and mortality 2.0–3.7 per 100,000 women) [2]. Although genetic predisposition and racial background might promote EC development, the most important EC-predisposing factors seem to be associated with health and lifestyle conditions (e.g., obesity, metabolic syndrome, diabetes, polycystic ovary syndrome, high estrogen levels) [3,4,5,6].

EC is classified into two subtypes with distinct clinical, pathological and molecular features. Commonly, type I ECs display a low grade (I or II) endometrioid morphology and are estrogen-dependent; thus, they are associated with a good prognosis. Type II ECs include non-endometrioid adenocarcinomas, serous clear cell, undifferentiated carcinomas and carcinosarcomas, usually hormone-receptor negative high-grade tumors, with poor prognosis [7].

Type I EC comprises the large majority of endometrial cancers, ~90%, while type II EC comprises ~10%. In type I EC, stage 1 is the most frequent [8]. In EC, 80% of patients are in the early stages, while 20% are in more advanced phases [9]. At stage I, the five-year survival rate is 95%, and the survival decreases dramatically to 14% for stage IV [10]. The identification of biomarkers at an early stage would lead to a prompt diagnosis, reduce inappropriate and invasive examinations and improve patient care and prognosis [11]. Patients in more advanced stages require more specific care, such as radio therapy or chemotherapy. For this purpose, identification of biomarkers at an early stage is important. Harsh treatment can be avoided for many patients when EC is discovered at an early stage [12].

Growing pieces of evidence showed that, by releasing soluble mediators and extracellular vesicles, EC cells are able to recruit stromal cell, immune system cells and macrophages [13,14], which, in turn, favor tumor growth.

Interestingly, inflammation can exert a double role in cancer development. On one hand, triggers of inflammation activate immune cells that, in turn, produce suppressive tumor growth cytokines [15]. Thus, inflammation can induce the response of the immune system against cancer. On the other hand, inflammation is involved in the promotion of tumorigenesis [16], for example, several cytokines such as IL-6 [17], TNFα [18], TGF-β [19], and IFNs [20] are involved in cancer development and progression.

Cytokines’ profiles are promising in the diagnosis and prognosis of pancreatic [21], colorectal [22] and ovarian cancers [23]. Multi-omics approaches allowed us to understand several tumor mechanisms and, of note, the identification of diagnostic and prognostic markers [24]. The proteomic approach allowed for the identification of several candidate biomarkers in the serum of EC patients, such as CLU, C1R, and SERPINC1 [25], SBSN [26], and PAK1 [27].

In this study, we used proximity extension assays (PEAs) for target biomarker analysis in serum from EC patients. Such technology uses high-multiplex matched pairs of antibodies, labelled with unique DNA oligonucleotides, which bind to their specific/relative proteins in the samples [28]. Several studies used PEA for targeted biomarker analysis in different types of cancers, such as esophageal squamous cell carcinomas [29], cervical cancer [30] and ovarian cancer [31]. The Olink PEA platform is a high-multiplex protein biomarker platform with a high-throughput capacity with high specificity and sensitivity in combination with minimal sample volume [32]. This powerful technology has been applied for the screening of proteins in blood [33,34], and on other matrices, e.g., cells [35], urine [36], peritoneal fluid [37], and exosomes [38]. Moreover, recently it has been applied to study immunological mechanisms in the multisystem inflammatory syndrome in children with COVID-19 [39].

Our study aims to apply the PEA technology to screen candidate serum protein markers of early-stage of type I EC.

## 2. Materials and Methods

### 2.1. Patients

For this study, a total of 88 women (44 suffering from EC and 44 non-EC controls) were recruited at the Institute for Maternal and Child Health—IRCCS “Burlo Garofolo” (Trieste, Italy) from 2019 to 2021. All EC patients had type I endometrioid adenocarcinomas at tumor stage 1. Type I EC comprises the large majority of endometrial cancers, ~90%, while the type II EC comprises ~10%. In type I EC, stage 1 is the most frequent. For this reason, in this study we focused on type I EC at stage 1 patients to identify candidate serum protein biomarkers.

All procedures complied with the Declaration of Helsinki and were approved by the Institute’s Technical and Scientific Committee. All patients signed informed consent forms. In Appendix A, we describe the clinical and pathological characteristics of the patients. The median age of patients was 67 years (Inter quartile range 55–71), with a minimum of 44 and a maximum of 81, while the median age of controls was 35 years (IQR 27–51), with a minimum of 22 and a maximum of 77 years. Controls were chosen excluding oncologic patients, human immunodeficiency virus (HIV), hepatitis B virus (HBV), hepatitis C virus (HCV) seropositive subjects, and patients with leiomyomas or adenomyosis. For EC cases, we ruled out women with other oncologic pathologies, human immunodeficiency virus (HIV), hepatitis B virus (HBV), hepatitis C virus (HCV) seropositive patients, and patients with leiomyomas or adenomyosis. We excluded control patients with benign tumors (myoma), chronic inflammatory disease (adenomyosis) or viral infections because these pathologies may influence the abundance of several proteins in serum and, consequently, affect the proteomic analysis.

### 2.2. Serum Sample Collection and the PEA

To obtain serum, blood was centrifuged at 5000 rcf × 5 min. Once obtained, the serum was preserved at −80 °C. Sera were shipped to Olink^®^ Proteomic (Dag Hammarskjölds väg 52B, SE-752 37 Uppsala, Sweden). In total, 40 µL of serum was used for PEA analysis in the Immuno-oncology panel and the Target 96 Oncology III panel with 96-wells, in which 92 oligonucleotide-labeled antibody probe pairs bind to their specific targeted proteins. The protein names, gene names, and abbreviations for the 92 proteins of the Immuno-oncology panel and Target 96 Oncology III panel are reported in Appendix A. The PEA technology includes three core steps. It starts with an overnight incubation of 16–22 h. The 92 antibody pairs, labelled with DNA oligonucleotides, bind to their respective protein in the samples. During the second step, we have 2 h of extension and amplification. Oligonucleotides that are brought into proximity hybridize, and are extended using a DNA polymerase. This newly created piece of DNA barcode is amplified by PCR. In the last step, we have 4.5 h of detection. The amount of each DNA barcode is quantified by microfluidic qPCR.

Negative Control for Olink Explore is also included in triplicate on each plate and consists of buffer run as a normal sample. These are used to monitor any background noise generated when DNA-tags come in close proximity without prior binding to the appropriate protein. The negative controls set the background levels for each protein assay and are used to calculate the limit of detection (LOD) and to assess the potential contamination of the assays. The Plate Control was another control included in triplicate on each plate. The median of the Plate Control triplicates is used to normalize each assay and compensate for the potential variation between runs and plates. Once the data were obtained from the plate reading, they were analyzed, including normalization and linearization, by protocols of the manufacturer. The protein level is expressed as NPX, Normalized Protein eXpression, an arbitrary unit in Log2 scale. It is calculated from Ct values, and data pre-processing (normalization) is performed to minimize both intra- and inter-assay variation. NPX data allow users to identify changes in individual protein levels across their sample set, and then use these data to establish protein signatures.

Olink Target 96 Immuno-oncology panels include proteins associated with biological functions linked to immune response and immuno-oncology diseases. The biomarkers in this panel include proteins involved in processes such as promotion and inhibition of tumor immunity, chemotaxis, vascular and tissue remodeling, apoptosis and cell killing and metabolism and autophagy. The Olink Target 96 oncology III panel comprises 92 human proteins that participate in biological mechanisms that are central to the initiation and progression of cancer, e.g., angiogenesis, cell communication, cellular metabolic processes, apoptosis, cell proliferation/differentiation, etc. In Appendix A, all the proteins that make up the two panels are reported. These panels do not focus on specific malignancies. The categorization of the proteins included in the panel was carried out via referral to widely used public-access bioinformatic databases, including Uniprot, Human Protein Atlas, Gene Ontology (GO) and DisGeNET.

### 2.3. Bioinformatic Analysis

Proteins used for statistical analysis for both the panels were analyzed by gProfiler [40] (https://biit.cs.ut.ee/gprofiler/gost) accessed on 4 July 2022 classification systems and categorized according to their: molecular function involvement, biological processes, and protein class. Pathway analysis was done by REACTOME tool [41] (https://reactome.org/PathwayBrowser/#TOOL=AT) accessed on 4 July 2022.

### 2.4. Statistical Analyses

Each of the two panels comprised 92 proteins. We first excluded all proteins with more than 25% values below the limit of detection (LOD). Olink suggests excluding assays in the range of less than 25–50% of samples above LOD (https://www.olink.com/faq/how-is-the-limit-of-detection-lod-estimated-and-handled/) accessed on 4 July 2022, but we adopted a more restrictive approach and chose to have at least 75% of samples above the LOD. After excluding these, for each of the remaining proteins, we calculated the median value, and the interquartile range. We carried out binary logistic regressions to study the association with the outcome. Before proceeding to the multivariate logistic regression, for each panel, we selected the proteins more strongly associated with the dependent variable, on the basis of the binary logistic regression *p*-value result. Our first approach was to carry out a first selection of proteins with a least absolute shrinkage and selection operator (LASSO) multivariate logistic regression approach, but the results were unfortunately not comparable with a traditional binary logistic regression approach. For each panel separately, the selected proteins were simultaneously considered in a multivariate logistic regression model. A downward selection was applied to exclude, one at a time, the proteins with the highest *p*-value, if *p* ≥ 0.05. We thus obtained two final predictive models which included only proteins significantly and simultaneously associated with the outcome. For each model, we reported the Pseudo-R-squared value, the Area under the Receiver Operating Characteristic (ROC) Curve (AUC), sensitivity and specificity. Finally, we decided to consider the two final models together in a multivariate logistic regression model. we hypothesize that improving the predictive models we might obtain a group of proteins that could be included in an ad hoc panel. Again, we adopted a stepdown procedure and obtained a third model.

## 3. Results

In the first panel—the Immuno-oncology panel—ten proteins had more than 25% values below the LOD (IL_1_alpha, FGF2, IL2, IL33, CD28, IL5, PTN, CXCL12, IL4, IL13) and were excluded from further analyses. Of the remaining proteins, median values and interquartile ranges are reported for cases and controls, as well as the odds ratios, 95% confidence intervals and *p*-values of the binary logistic regression (Table 1).

The binary logistic regression analyses allowed us to identify the proteins more strongly associated with the outcome. For the first panel, we selected 20 proteins with a binary logistic regression *p*-value below 0.01. These 20 proteins were considered together in a multivariate logistic regression model (Table 2). After applying a downward selection as described in the Methods section, four proteins remain, significantly and simultaneously associated with the outcome (Table 3).

This model has a Pseudo R-squared = 0.605, an AUC = 95.4% (95% CI 91.5–99.3%), reaching a sensitivity of 97.67% with a specificity of 74.42% (Table 4 and Figure 1 and Figure 2). For regression coefficients reported in Table 3 and predicted probability cut points reported in Table 4, the following model will identify cases and controls with the specified sensitivity and specificity:

Predicted probability = 1/(1 + exp(–(–87.09041 − 3.352554 × Gal-9 + 9.833984 × Gal-1 + 5.496387 × MMP7 − 3.052633 × FASLG))).

In the second panel—the Target 96 Oncology III panel—there were 19 proteins with more than 25% values below the limit of detection (TBL1X, IL17F, TPMT, KLK4, NT5C3A, GAMT, HEXA, TNFAIP8, AIF1, CNPY2, SEPT9, CDC27, CXCL14, LAP3, SPINK4, YTHDF3, ACTN4, GGA1, TPT1) so they were excluded from further analyses. Of the remaining proteins, in Table 5 we report median values and interquartile ranges for cases and controls, as well as the odds ratios, 95% confidence intervals and *p*-values of the binary logistic regression.

With the results of the binary logistic regression, for the second panel, we selected 24 proteins with a binary logistic regression *p*-value below 0.02, as we only had 11 proteins with a *p*-value below 0.01 (Table 6). These 24 proteins were considered together in the multivariate logistic regression model. After applying a downward selection, we were left with five proteins, significantly and simultaneously associated with the outcome (Table 7).

This model has a Pseudo R-squared = 0.436, an AUC = 88.9% (82.1–95.6%), reaching a sensitivity of 95.45% with a specificity of 69.77% (Table 8 and Figure 3 and Figure 4). For the regression coefficients reported in Table 7 and predicted probability cut points reported in Table 8, the following model identified cases and controls with the specified sensitivity and specificity:

Predicted probability = 1/(1 + exp(−(−21.76806 + 0.8618858 × CDHR2 + 3.415408 × NCS1 + 0.6490679 × MLN − 2.915442 × FLT3 − 1.256405 × COL9A1))).

The third model was generated by considering in a multivariate logistic regression all proteins included in the two final models, i.e., Gal-1, Gal-9, MMP7. FASLG, CDHR2, NCS1, MLN, FLT3 and COL9A1 (Table 9). After the stepdown procedure, the final model included all variables previously included in the immune-oncology final model, plus COL9A1 (Table 10). This model has a Pseudo R-squared = 0.691, an AUC = 96.9% (93.9–99.9%), reaching a sensitivity of 97.67% with a specificity of 83.72% (Table 11, Figure 5 and Figure 6). Regression coefficients are reported in Table 10. Predicted probability cut points for specificity higher than sensitivity are reported in Table 11. The predicted probability can be calculated from the following model:

Predicted probability = 1/(1 + exp(−(−121.4969 − 4.713017 × Gal-9 + 11.1979 × Gal-1 + 9.248928 × MMP7 − 4.163016 × FASLG − 2.687621 × COL9A1))).

### Bioinformatic Analysis

We used gProfiler as the classification tool for proteomic enrichment data analysis. For the Immuno-oncology panel, proteins (Figure 7) are classified into groups according to their molecular function, biological processes, and protein classes. Regarding molecular function, proteins were categorized into: cytokine receptor binding, cytokine activity, receptor-ligand activity, signaling receptor activator activity, signaling receptor regulator activity, signaling receptor binding, chemokine activity, and chemokine receptor binding. For biological processes, proteins were categorized into: immune response, immune system process, positive regulation of immune system process, cell surface receptor signaling pathway, regulation of immune system process, cytokine-mediated signaling pathway, response to cytokine, and cellular response to cytokine stimulus. Contrastingly, for protein class, proteins were categorized into: extracellular region, external side of plasma membrane, extracellular space, cell surface, side of membrane, cell periphery, plasma membrane, and integral component of plasma membrane. Reactome tool grouped these proteins into eight pathways: chemokine receptors bind chemokines, interleukin-10 signaling, cytokine signaling in immune system, immune system, signaling by interleukins, peptide ligand-binding receptors, TNFR2 non-canonical NF-kB pathway, class A/1 (Rhodopsin-like receptors).

The same analysis was performed for the Target 96 Oncology III panel (Figure 8). Proteins were classified into groups according to their molecular function, biological processes, and protein class. The molecular function categories were: protein binding. The biological processes were: negative regulation of endothelial cell proliferation. The protein class categories were: extracellular region, extracellular space, vesicle, extracellular exosome, extracellular vesicle, extracellular organelle, and extracellular membrane-bounded organelle. Reactome tool analysis classified these proteins into four pathways: CLEC7A/inflammasome pathway, Defective CSF2RA causes SMDP4, and Defective CSF2RB causes SMDP5, Innate Immune System.

## 4. Discussion

Biomarkers play a key role in oncological applications, including diagnosis of the disease, prognosis and determination of personalized treatment [42]. High throughput technology of TMT LC\MS-MS allowed for the identification and quantification of several candidate biomarkers in a large cohort of patients [43]. In recent years, there has been a great effort to identify and validate biomarkers in EC, but, until now, no candidate biomarker has reached the clinical stage [43,44].

To identify early candidate biomarkers in EC, we exploited for the first time PEA technology in targeted proteomics, which had already been used successfully for identification of biomarkers in several pathologies. From the 92 proteins of the Immuno-oncologic panel, only 20 were selected with a binary logistic regression *p*-value below 0.01. A multivariate logistic regression analysis based on four proteins (Gal-9, Gal-1, MMP7, FASLG) allowed us to separate cases from controls with an AUC = 95.4%. From the Target 96 Oncology III, only 24 proteins were selected with a binary logistic regression *p*-value below 0.02. A multivariate logistic regression analysis based on five proteins (CDHR2, NCS1, MLN, FLT3, COL9A1) allowed us to separate cases from controls with an AUC = 88.9%. According to these results, the performance of the Immuno-oncologic panel was better than the Target 96 Oncology III panel. To further improve the model, we performed a multivariate logistic regression, including all proteins from the first (Gal-1, Gal-9, MMP7, FASLG) and one from the second model (COL9A1) obtaining an AUC = 96.9%.

Thanks to PEA technology, several candidate biomarkers were identified in different pathologies, such as FABP2, FGF5, LPL, and LTA in coronary artery disease (CAD) pathogenesis [45]; ITGAV, EpCAM, IL18, SLAMF7 and IL8 were identified as biomarkers in inflammatory bowel disease (IBD) [46]; and ten candidate biomarkers, including CHIT1, SMOC2, MMP-10, LDLR, CD200, EIF4EBP1, ALCAM, RGMB, tPA and STAMBP, were identified in early Alzheimer’s disease [47]. In endometriosis, PEA technology identified seven proteins up-regulated (IL-6, IL-8, CCL 19, SCF, VEGF-D, IL-6RA, MIA9) and ten proteins down-regulated (ICOSLG, EGFR, SELE, ErbB2/HER2, IL-6RA, VEGFR-2, Flt3L, CXCL10, HE4, FR-alpha). Moreover, in endometriosis patients, dysregulation of these proteins has been reported to induce immune response modulation, angiogenesis, cell proliferation, cell adhesion and inhibition of apoptosis [38].

In this work, we identified five proteins, namely, Gal-1, Gal-9, MMP7, FASLG, and COL9A1, that based on their known function in endometrial and other cancers might represent useful early-stage EC biomarkers, upon a validation phase.

Indeed, Extracellular Gal-9 induces apoptosis of effector T-cells through mucin domain-containing molecule 3 (Tim-3) [48]. This protein induces the differentiation and suppressive activity of regulatory T-cells [49]. In their work, Chuan-xia Zhang et al. highlighted a novel mechanism involving Gal-9 in creating an immune-suppressive microenvironment, which favors tumor progression [50].

Gal-1 is a small lectin that binds beta-galactoside and a wide array of complex carbohydrates. This protein acts as an immunosuppressive molecule and is expressed by different types of cancer cells [51]. Once secreted, Gal-1 binds to the glycosylated receptor of immune cells, leading to their inhibition and consequently to the immune escape of cancer cells [52]. In EC, Mylonas and colleagues performed an immunohistochemistry study of Gal-1 and Gal-9, finding a correlation between these proteins and EC clinicopathological features. [53] Indeed, high expression of Gal-1 is associated with poor prognosis, while high expression of Gal-9 is associated with early pathological changes. In addition, Gal-1 also correlates with lymphangiosis, a poor prognostic marker in EC [54].

MMP7 is a small enzyme that degrades several types of galectins, casein and fibronectin [55]. MMP7 promotes tumor cell invasion and migration, digesting the extracellular matrix (ECM) and components of cell surface proteins [56]. This protein is a promising diagnostic and prognostic biomarker of pancreatic cancer [57] and bladder cancer [58]. Its biochemical characteristics make MMP7 a potential target in several types of cancers [56]. Downregulation of MMP7 leads to a reduced proliferation and migration of tumor cells in gallbladder cancer [59] and reduces the cisplatin resistance in NCSLC cells [60]. In EC, high expression of MMP7 correlates with higher lymph node invasion [61] and increased risk of metastasis [62]. These data are supported also by in vitro assays (Misugi et al.), confirming that increased expression of MMP-7 in high-grade ECs may be correlated with tumor invasion and the protein may be a prognostic marker in EC [63].

COL9A1 is a structural component of hyaline cartilage and vitreous of the eye; the COL9A1 gene localizes in chromosome 16q13 [64]. Several studies correlates COL9A1 with breast cancer [65] and oral squamous cell carcinoma [66]. In EC, computational analysis of RNAseq data deposited in the TCGA database show that COL9A1 expression is increased both in primary and metastatic tumor (https://www.ncbi.nlm.nih.gov/pmc/articles/PMC7173926/) accessed on 4 July 2022.

Lastly, FASLG binds to TNFRSF6/FAS receptor inducing apoptosis [67]. This protein is important to maintain the immune homeostasis and for the elimination of cancer cells [68]. Epithelial mesenchymal transition (EMT) and stiffness make cancer cells more aggressive by excessive production and secretion of FASLG [69]. McGlorthan et al. described a possible mechanism explaining how progesterone and calcitriol induce apoptosis in EC, relying on the induction of FasL, Fas, and FADD expression, which, in turn, activates the caspase-8 pathway [70]. Accordingly, a genetic study showed that the homozygous CC variant of FASL −844 T>C polymorphism confers protection against EC [71].

The Reactome analysis of our five candidate proteins also identified several pathways related with the immunity system. Of them, chemokines and chemokine receptors not only take part in immune regulation but also play a key role in tumor development. Moreover, chemokines and chemokine receptors are related with angiogenesis, metastasis, drug resistance, and immunity of breast cancer [72]. For example, IL-10 plays a key role on the regulation of several genes in gastric cancer cells involved in cell proliferation and migration [73]. Interestingly, cytokines are small proteins that play an important role in cellular functions, such as proliferation, differentiation, and survival, as well as the response to pathogens. These proteins induce the activation of the JAK-STAT pathway, which is fundamental in the regulation of immune system and tumor surveillance [74].

Altogether, in our study, we found a panel of proteins whose secretion in blood correlates with early EC and can be exploited as a diagnostic biomarker upon further validation, although we know that there are some limitations. First, since it was very difficult for us to find controls with the same age as the EC patients, the controls are slightly younger than the EC patients. We acknowledge this is a limitation and that levels of some proteins might change with age. The next step will be the validation of these results with age-matched cases and controls.

Another limit of our study is the small number of patients, which does not allow us to generalize to the overall population. Even if the results are satisfactory, more studies are certainly required to confirm and consolidate these findings.

## 5. Conclusions

In conclusion, by combining proteins from the Immuno-oncology panel and the Target 96 Oncology III panel, we were able to generate an algorithm that was able to discriminate early EC type I patients from controls with high specificity and sensitivity thanks to the analysis of Gal-1, Gal-9, MMP7, COL9A1, and FASLG serum levels. Although Gal-1, Gal-9, MMP7, COL9A1, and FASLG are overexpressed in different kinds of cancers, the analysis of their serum levels allows one to discriminate between healthy controls and woman affected by type I EC, and combining this analysis with a typical clinical manifestation of EC, such us bleeding and pelvic pain, might help early EC diagnosis, avoiding invasive diagnostic techniques.

## Figures and Tables

**Figure 1 biomedicines-10-01857-f001:**
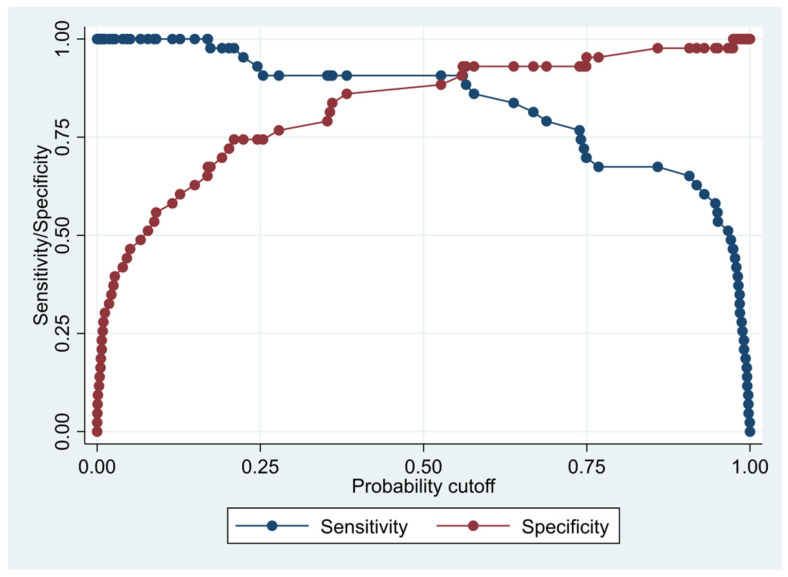
Sensitivity and specificity plot of the final multivariate logistic regression model based on cytokines from the immune-oncology panel.

**Figure 2 biomedicines-10-01857-f002:**
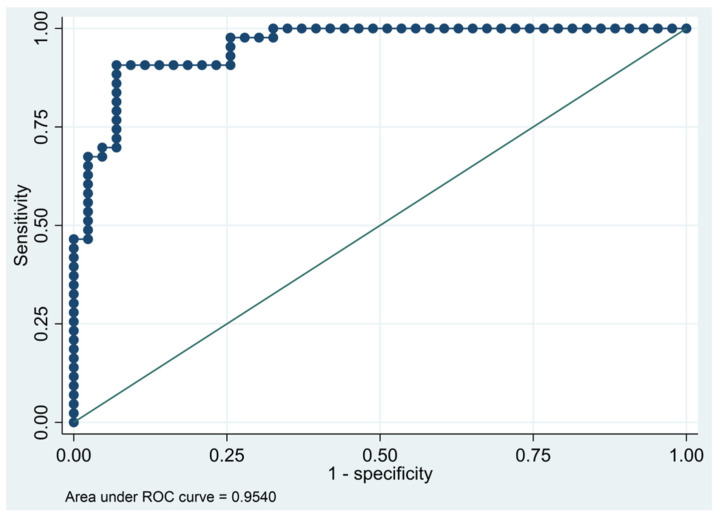
Receiver operating characteristics curve of the final multivariate logistic regression model based on cytokines from the immune-oncology panel.

**Figure 3 biomedicines-10-01857-f003:**
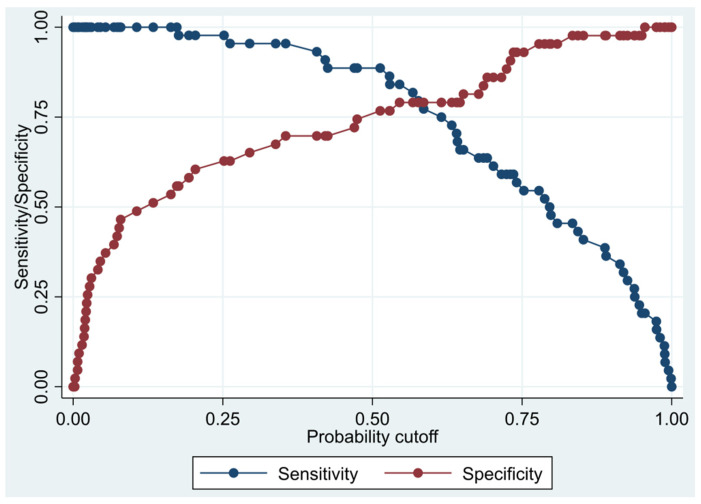
Sensitivity and specificity plot of the final multivariate logistic regression model based on cytokines from the oncology panel.

**Figure 4 biomedicines-10-01857-f004:**
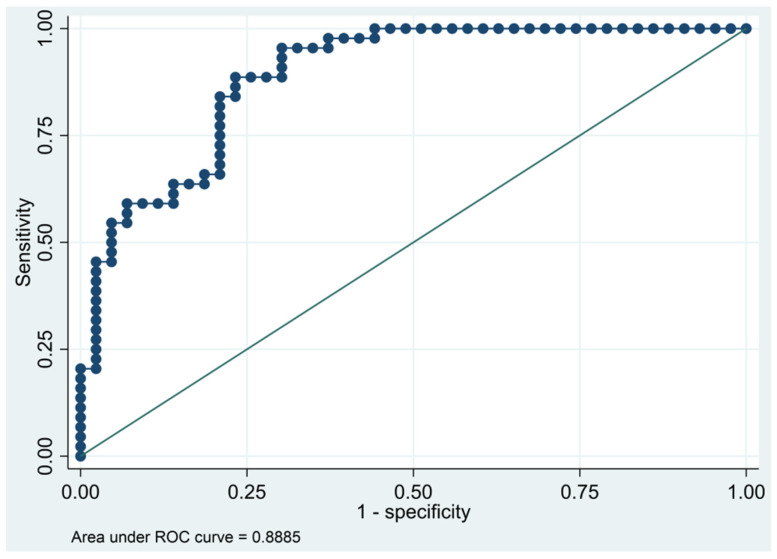
Receiver operating characteristics curve of the final multivariate logistic regression model based on cytokines from the oncology panel.

**Figure 5 biomedicines-10-01857-f005:**
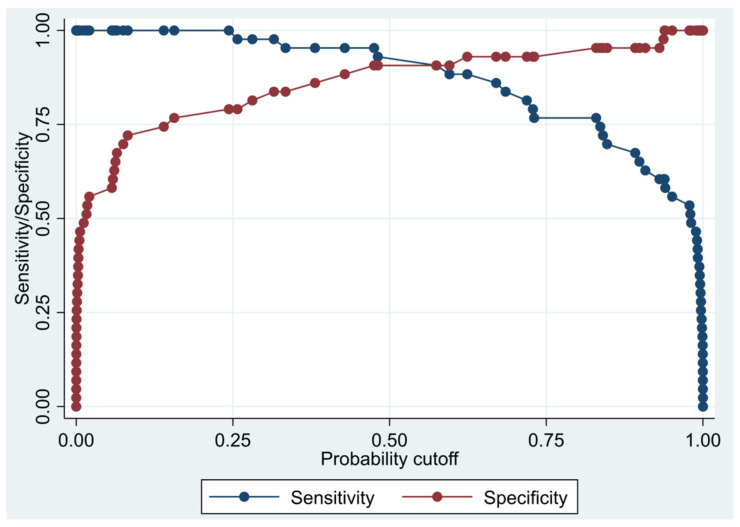
Sensitivity and specificity plot of the final multivariate logistic regression model based on cytokines resulting from the two final models based on both the immune-oncology and the oncology panel.

**Figure 6 biomedicines-10-01857-f006:**
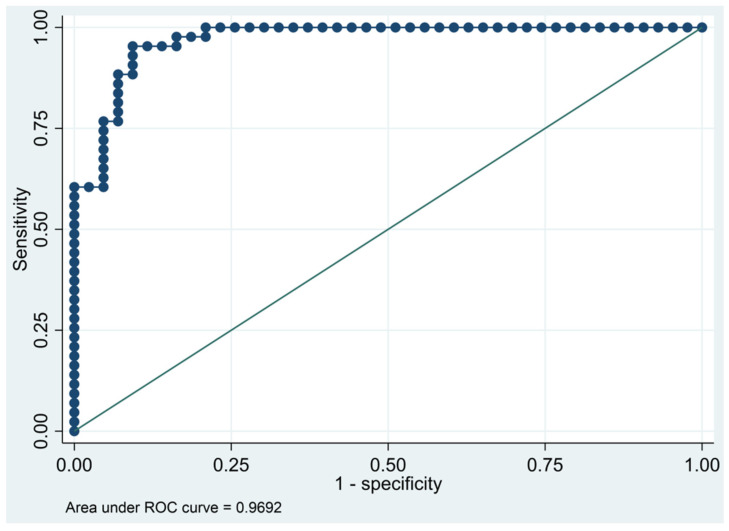
Receiver operating characteristics curve of the final multivariate logistic regression model based on cytokines resulting from the two final models based on both the immune-oncology and the oncology panel.

**Figure 7 biomedicines-10-01857-f007:**
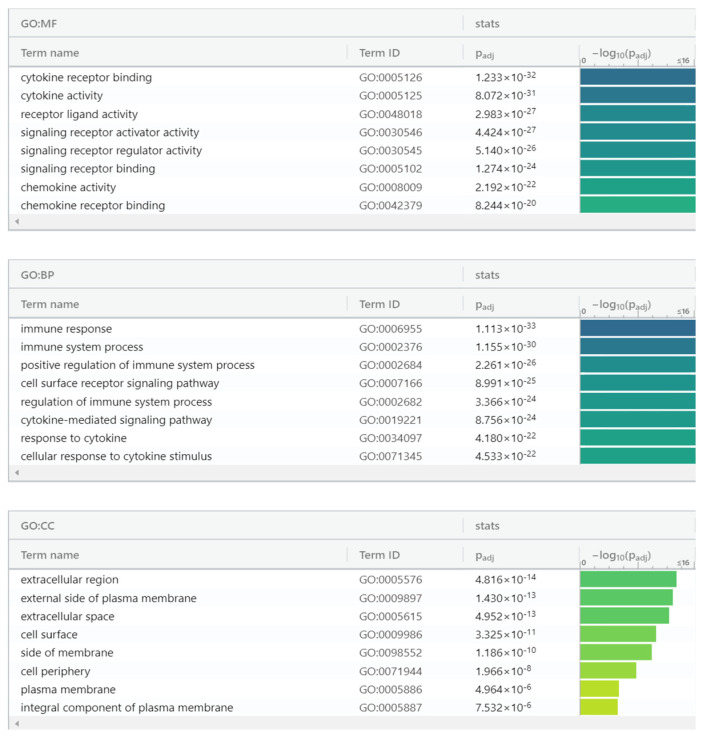
gProfiler classification of Immuno-oncologic panel proteins in the EC serum according to their molecular function, biological processes, cellular component.

**Figure 8 biomedicines-10-01857-f008:**
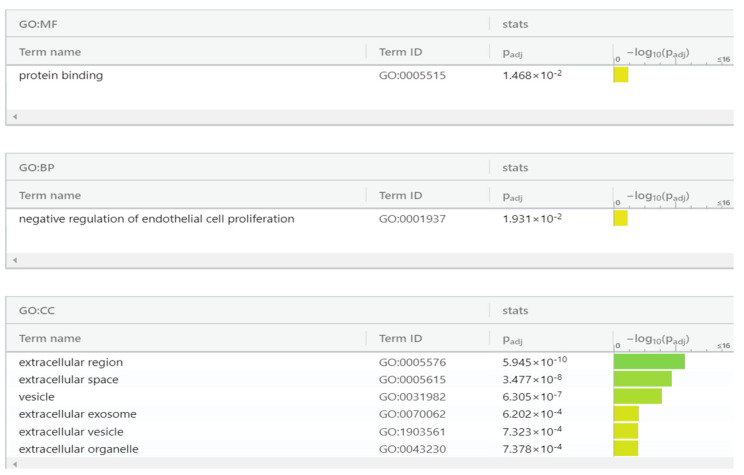
gProfiler classification of Oncology III panel proteins in the EC serum according to their molecular function, biological processes, cellular component.

**Table 1 biomedicines-10-01857-t001:** Immuno-oncology panel: result of descriptive analysis, and binary logistic regression against the outcome.

Immuno-Oncology	Cases	Controls	Binary Logistic Regression
Proteins	Median (IQR)	Median (IQR)	OR (95% CI); *p*-Value
IL8	6.044 (5.629–6.517)	5.66 (5.336–6.106)	1.291 (0.817–2.039); 0.274
TNFRSF9	6.485 (6.147–6.626)	6.043 (5.905–6.246)	17.67 (3.931–79.432); 0.000
TIE2	7.485 (7.269–7.708)	7.487 (7.229–7.572)	1.45 (0.302–6.968); 0.643
MCP-3	2.057 (1.847–2.303)	1.762 (1.402–2.24)	2.264 (1.031–4.975); 0.042
CD40-L	8.018 (7.329–8.390)	7.889 (6.814–8.276)	1.195 (0.808–1.766); 0.372
CD244	6.385 (6.192–6.586)	6.299 (6.128–6.476)	1.542 (0.395–6.020); 0.533
EGF	9.054 (7.886–9.7600)	9.146 (7.360–9.413)	1.164 (0.829–1.634); 0.381
ANGPT1	9.769 (9.487–9.968)	9.811 (9.624–9.877)	0.588 (0.129–2.691); 0.494
IL7	5.952 (5.601–6.553)	6.095 (5.696–6.534)	0.889 (0.450–1.756); 0.735
PGF	8.295 (8.186–8.506)	8.051 (7.945–8.245)	16.516 (3.089–88.316); 0.001
IL6	3.539 (3.022–4.38)	2.462 (1.888–3.612)	1.850 (1.263–2.708); 0.002
ADGRG1	1.796 (1.570–2.764)	1.627 (1.364–1.878)	3.126 (1.410–6.930); 0.005
MCP-1	11.488 (11.293–11.832)	11.147 (10.71–11.44)	4.318 (1.689–11.038); 0.002
CRTAM	5.762 (5.335–5.985)	5.473 (5.131–5.894)	2.086 (0.928–4.692); 0.075
CXCL11	7.766 (6.873–8.155)	7.388 (6.871–7.799)	1.568 (0.864–2.845); 0.139
MCP-4	11.037 (10.626–11.529)	10.638 (10.213–10.873)	2.116 (1.073–4.176); 0.031
TRAIL	8.094 (7.763–8.288)	7.870 (7.551–8.133)	3.787 (1.134–12.65); 0.030
CXCL9	7.049 (6.571–7.811)	6.054 (5.777–6.732)	3.690 (1.932–7.049); 0.000
CD8A	9.223 (8.884–9.765)	9.497 (9.191–9.789)	0.642 (0.321–1.286); 0.211
CAIX	4.500 (4.237–4.932)	4.241 (3.967–4.693)	3.354 (1.274–8.826); 0.014
MUC-16	2.053 (1.785–2.757)	2.034 (1.758–2.585)	1.118 (0.626–1.994); 0.707
ADA	5.288 (4.916–5.892)	5.109 (4.856–5.507)	1.755 (0.903–3.408); 0.097
CD4	3.475 (3.261–3.727)	3.279 (3.094–3.446)	8.973 (1.963–41.013); 0.005
NOS3	2.06 (1.689–2.355)	1.782 (1.580–1.988)	3.899 (1.392–10.92); 0.010
Gal-9	7.903 (7.443–8.121)	7.358 (7.162–7.523)	15.109 (4.003–57.025); 0.000
VEGFR-2	8.229 (8.034–8.477)	8.24 (8.119–8.532)	0.542 (0.137–2.143); 0.383
CD40	9.506 (9.302–9.752)	9.302 (9.059–9.508)	4.179 (1.224–14.262); 0.022
IL18	8.915 (8.372–9.222)	8.539 (8.198–8.891)	3.243 (1.311–8.024); 0.011
GZMH	3.875 (3.185–4.346)	3.207 (2.837–4.018)	1.885 (1.094–3.249); 0.022
KIR3DL1	1.930 (0.995–2.674)	1.789 (0.567–2.509)	1.098 (0.753–1.601); 0.628
LAP TGF-beta-1	9.665 (9.136–9.978)	9.599 (9.333–9.889)	1.080 (0.429–2.716); 0.871
CXCL1	9.135 (8.801–9.48)	8.988 (8.716–9.405)	1.239 (0.532–2.888); 0.619
TNFSF14	6.359 (5.559–6.991)	6.250 (5.312–7.117)	1.112 (0.740–1.672); 0.609
TWEAK	8.485 (8.272–8.749)	8.600 (8.291–8.790)	0.505 (0.147–1.741); 0.279
PDGF subunit B	10.638 (10.589–10.686)	10.636 (10.595–10.674)	1.665 (0.009–302.191); 0.848
PDCD1	4.551 (4.369–4.836)	4.682 (4.410–4.853)	0.640 (0.188–2.175); 0.475
FASLG	6.765 (6.439–6.969)	7.175 (6.813–7.377)	0.171 (0.058–0.505); 0.001
CCL19	10.521 (9.986–11.038)	10.327 (9.858–10.934)	1.305 (0.857–1.988); 0.215
MCP-2	8.183 (7.845–8.681)	8.031 (7.578–8.473)	1.880 (0.904–3.909); 0.091
CCL4	6.813 (6.568–7.372)	6.626 (6.289–6.890)	3.144 (1.268–7.794); 0.013
IL15	5.212 (4.91–5.463)	5.029 (4.797–5.233)	4.885 (1.306–18.273); 0.018
Gal-1	6.801 (6.594–6.966)	6.455 (6.234–6.597)	353.644 (27.173–4602.582); 0.000
PD-L1	5.647 (5.414–5.886)	5.497 (5.374–5.740)	4.928 (0.994–24.444); 0.051
CD27	8.181 (7.874–8.406)	7.902 (7.697–8.119)	6.518 (1.706–24.899); 0.006
CXCL5	12.024 (11.577–12.602)	11.966 (11.315–12.334)	1.246 (0.687–2.260); 0.469
HGF	9.226 (8.754–9.662)	9.115 (8.600–9.356)	2.108 (0.977–4.548); 0.057
GZMA	7.127 (6.771–7.371)	6.933 (6.514–7.267)	2.623 (0.952–7.226); 0.062
HO-1	12.008 (11.764–12.304)	11.886 (11.619–12.107)	2.405 (0.865–6.693); 0.093
CX3CL1	4.005 (3.683–4.300)	3.842 (3.615–4.053)	3.479 (0.971–12.459); 0.055
CXCL10	9.027 (8.522–9.557)	8.389 (7.868–8.740)	2.746 (1.454–5.186); 0.002
CD70	3.865 (3.46–4.199)	3.511 (3.26–3.760)	3.926 (1.433–10.756); 0.008
IL10	3.752 (3.312–4.120)	3.383 (3.105–3.858)	1.877 (0.994–3.543); 0.052
TNFRSF12A	4.851 (4.528–5.126)	4.456 (4.168–4.656)	11.654 (3.29–41.277); 0.000
CCL23	10.191 (9.773–10.574)	10.088 (9.936–10.469)	0.775 (0.302–1.991); 0.597
CD5	6.029 (5.763–6.276)	5.816 (5.612–6.044)	2.649 (0.892–7.865); 0.079
CCL3	6.250 (5.928–6.685)	5.864 (5.431–6.112)	4.251 (1.715–10.538); 0.002
MMP7	12.696 (12.569–12.849)	12.256 (12.061–12.455)	481.625 (36.262–6396.808); 0.000
ARG1	5.561 (5.000–6.640)	5.198 (4.880–6.030)	1.634 (1.002–2.666); 0.049
NCR1	3.714 (3.431–3.922)	3.509 (3.253–3.769)	2.378 (0.725–7.804); 0.153
DCN	4.535 (4.37–4.725)	4.315 (4.206–4.437)	111.093 (8.317–1483.879); 0.000
TNFRSF21	7.544 (7.398–7.662)	7.522 (7.373–7.628)	2.68 (0.361–19.901); 0.335
TNFRSF4	6.205 (5.943–6.493)	5.912 (5.759–6.194)	6.216 (1.657–23.316); 0.007
MIC-A/B	4.905 (4.395–5.533)	4.978 (4.537–5.412)	0.842 (0.590–1.200); 0.340
CCL17	10.277 (9.562–10.915)	10.105 (9.632–10.604)	1.377 (0.812–2.335); 0.235
ANGPT2	5.975 (5.62–6.302)	5.885 (5.634–6.270)	2.031 (0.675–6.116); 0.208
IFN-gamma	6.359 (5.913–6.877)	5.964 (5.427–6.330)	1.471 (0.931–2.324); 0.098
LAMP3	4.745 (4.284–5.43)	4.358 (4.113–4.677)	2.183 (1.12–4.255); 0.022
CASP-8	5.047 (4.103–6.314)	3.851 (3.396–4.414)	2.019 (1.338–3.048); 0.001
ICOSLG	5.940 (5.659–6.089)	5.983 (5.875–6.218)	0.608 (0.175–2.111); 0.433
MMP12	7.046 (6.772–7.771)	6.780 (6.33–7.309)	2.181 (1.171–4.063); 0.014
CXCL13	8.299 (8.071–8.524)	8.269 (7.877–8.605)	0.698 (0.307–1.588); 0.392
PD-L2	3.200 (2.792–3.464)	3.000 (2.794–3.283)	1.953 (0.613–6.22); 0.258
VEGFA	9.720 (9.345–10.084)	9.558 (9.332–9.887)	1.768 (0.723–4.326); 0.212
LAG3	4.523 (4.258–4.77)	4.462 (4.112–4.672)	4.164 (1.31–13.237); 0.016
IL12RB1	1.888 (1.711–2.114)	1.725 (1.565–1.934)	9.144 (1.67–50.067); 0.011
CCL20	5.710 (5.477–6.405)	5.499 (5.049–5.740)	2.036 (1.119–3.705); 0.020
TNF	3.914 (3.620–4.217)	3.831 (3.571–4.143)	1.515 (0.623–3.679); 0.359
KLRD1	6.55 (6.182–6.934)	5.953 (5.63–6.409)	5.074 (2.083–12.36); 0
GZMB	2.507 (2.139–2.804)	2.410 (2.135–2.804)	1.462 (0.623–3.429); 0.383
CD83	2.814 (2.665–3.039)	2.671 (2.523–2.903)	5.185 (1.065–25.239); 0.042
IL12	7.326 (6.955–7.723)	7.135 (6.782–7.510)	2.032 (0.928–4.447); 0.076
CSF-1	10.421 (10.227–10.561)	10.270 (10.083–10.440)	4.818 (0.927–25.049); 0.062

Note: IQR = Interquartile Range (25th and 75th centiles); OR = Odds Ratio; CI = Confidence Interval.

**Table 2 biomedicines-10-01857-t002:** Multivariate logistic regression model including all variables from the immune-oncology panel with *p* < 0.01 at binary logistic regression.

Protein	OR (95% CI)	*p*-Value
TNFRSF9	13.785 (0.014–1.38 × 10^4^)	0.457
CXCL9	2.164 (0.198–23.641)	0.527
Gal-9	5.09 × 10^−5^ (6.36 × 10^−10^–4.073)	0.086
Gal-1	2.82 × 10^8^ (93.752–8.50 × 10^14^)	0.011
TNFRSF12A	0.245 (0.003–20.627)	0.534
MMP7	7348.889 (3.198–1.69 × 10^7^)	0.024
DCN	0.472 (2.88 × 10^−5^–7731.066)	0.880
KLRD1	0.644 (0.015–27.749)	0.819
PGF	0.001 (1.75 × 10^−8^–20.145)	0.163
IL6	2.910 (0.509–16.642)	0.230
ADGRG1	2.162 (0.156–29.909)	0.565
MCP-1	0.082 (0.001–7.585)	0.279
CD4	638.942 (0.106–3.84 × 10^6^)	0.146
FASLG	2.32 × 10^−4^ (8.23 × 10^−8^–0.655)	0.039
CD27	1.740 (0.011–268.23)	0.829
CXCL10	1.066 (0.073–15.616)	0.963
CD70	5.843 (0.194–175.917)	0.310
CCL3	2.238 (0.128–39.183)	0.581
TNFRSF4	1.107 (0–2636.098)	0.979
CASP-8	0.791 (0.184–3.402)	0.753

Note: OR = Odds Ratio; CI = Confidence Interval.

**Table 3 biomedicines-10-01857-t003:** Immuno-oncology panel: result of the stepdown procedure applied to the saturated multivariate logistic regression model.

	*p*-Value	OR (95% CI)	Coefficient (95% CI)
Gal-9	0.025	0.035 (0.002–0.652)	−3.352554 (−6.277541–−0.4275675)
Gal-1	0.001	18657.14 (60.882–5717447)	9.833984 (4.108936–15.55903)
MMP7	0.001	243.809 (9.623–6177.328)	5.496387 (2.264132–8.728641)
FASLG	0.003	0.047 (0.006–0.358)	−3.052633 (−5.07917–−1.026096)
Constant	0.000	1.5 × 10^−38^ (8.12 × 10^−59^–2.79 × 10^−18^)	−87.09041 (−133.7588–−40.42205)

Note: OR = Odds Ratio; CI = Confidence Interval; Coefficient = logistic regression model coefficient.

**Table 4 biomedicines-10-01857-t004:** Immuno-oncology panel: sensitivity/specificity plot for the resulting multivariate logistic regression model, and for sensitivity levels higher than specificity levels.

Predicted Probability Cut Point	Sensitivity	Specificity
≥0.1696234	100.00%	67.44%
≥0.2097338	97.67%	74.42%

**Table 5 biomedicines-10-01857-t005:** Oncology panel: result of descriptive analysis, and binary logistic regression against the outcome.

Oncology	Cases	Controls	Binary Logistic Regression
Proteins	Median (IQR)	Median (IQR)	OR (95% CI): *p*-Value
CD22	5.633 (5.371–5.891)	5.945 (5.565–6.088)	0.355 (0.126–1.003); 0.051
INPP1	1.444 (1.124–1.907)	1.144 (0.934–1.456)	1.607 (0.834–3.095); 0.156
ALPP	6.802 (5.775–7.713)	5.836 (4.828–6.321)	1.637 (1.172–2.287); 0.004
CGB3	5.052 (4.473–5.432)	3.732 (2.946–4.752)	2.642 (1.627–4.291); 0.000
NAMPT	2.006 (1.525–3.161)	1.557 (1.194–2.358)	1.334 (0.957–1.859); 0.089
VMO1	2.266 (1.916–2.404)	1.919 (1.522–2.236)	2.874 (1.156–7.142); 0.023
IFNGR2	4.891 (4.646–5.139)	4.775 (4.582–5.196)	1.478 (0.679–3.217); 0.325
ERP44	5.901 (5.621–6.108)	5.612 (5.318–5.899)	3.633 (1.156–11.418); 0.027
CBLN4	3.444 (3.298–3.680)	3.698 (3.314–3.977)	0.343 (0.120–0.985); 0.047
ACAA1	2.427 (1.879–3.125)	1.923 (1.734–2.484)	1.704 (1.033–2.808); 0.037
S100A16	2.739 (2.354–3.118)	2.937 (2.509–3.202)	0.732 (0.405–1.326); 0.304
PSPN	2.927 (2.576–3.874)	3.335 (2.624–3.773)	0.885 (0.523–1.498); 0.649
DCTPP1	4.963 (4.706–5.173)	4.683 (4.342–4.924)	6.91 (2.082–22.935); 0.002
MANSC1	5.962 (5.765–6.178)	6.081 (5.803–6.264)	0.843 (0.226–3.149); 0.800
GFER	3.559 (3.172–4.142)	3.329 (3.056–3.578)	2.775 (1.291–5.967); 0.009
RP2	1.799 (1.344–2.267)	1.534 (1.200–1.905)	1.886 (0.988–3.603); 0.055
JCHAIN	3.420 (3.081–3.917)	3.483 (3.202–3.705)	1.173 (0.474–2.903); 0.730
RAB6A	4.359 (3.629–5.391)	4.064 (3.434–4.728)	1.211 (0.879–1.669); 0.242
C1QA	6.135 (6.007–6.325)	6.205 (5.946–6.351)	1.273 (0.259–6.243); 0.766
AKR1B1	3.743 (3.193–4.232)	3.273 (2.876–4.084)	1.204 (0.763–1.902); 0.425
SCG2	3.605 (3.291–3.910)	3.677 (3.424–3.875)	0.805 (0.286–2.268); 0.681
RFNG	2.5100 (2.276–2.715)	2.280 (2.130–2.463)	9.739 (2.099–45.182); 0.004
MLN	3.789 (2.327–4.461)	2.639 (1.850–3.436)	1.771 (1.210–2.593); 0.003
ARHGAP25	1.381 (0.660–2.393)	1.111 (0.614–1.910)	1.283 (0.848–1.942); 0.238
IL1B	1.013 (0.548–1.575)	0.638 (0.309–1.319)	1.207 (0.844–1.726); 0.304
CCT5	0.806 (0.558–1.347)	0.492 (0.387–0.782)	3.433 (1.389–8.483); 0.008
CASP2	1.619 (1.216–2.204)	1.465 (0.942–2.340)	0.987 (0.686–1.420); 0.943
ELOA	1.234 (0.766–2.042)	0.905 (0.584–1.308)	1.384 (0.933–2.052); 0.106
NCS1	8.279 (8.000–8.556)	7.923 (7.704–8.144)	9.272 (2.593–33.153); 0.001
LSP1	4.291 (3.626–4.770)	4.224 (3.446–4.731)	1.198 (0.676–2.126); 0.536
AFP	5.767 (5.330–6.316)	5.373 (4.979–5.866)	2.166 (1.133–4.141); 0.019
GOPC	1.306 (0.968–1.783)	1.042 (0.833–1.247)	2.382 (1.071–5.302); 0.033
USO1	2.507 (1.866–3.311)	2.011 (1.725–2.592)	1.705 (1.067–2.723); 0.026
AIMP1	2.507 (1.830–3.010)	1.935 (1.625–2.538)	1.513 (0.925–2.476); 0.099
SCGN	2.439 (2.033–2.646)	2.173 (1.948–2.525)	1.858 (0.801–4.311); 0.149
TXNDC15	6.925 (6.815–7.071)	6.932 (6.830–7.047)	1.132 (0.142–9.049); 0.907
ICAM5	5.189 (4.957–5.424)	5.321 (4.973–5.628)	0.731 (0.254–2.110); 0.563
FUS	3.247 (2.068–4.311)	2.673 (1.860–3.261)	1.328 (0.943–1.871); 0.104
PTP4A1	1.073 (0.871–1.364)	1.078 (0.838–1.420)	1.105 (0.457–2.670); 0.825
FOXO3	1.495 (1.120–1.875)	1.126 (0.885–1.530)	1.709 (0.928–3.150); 0.086
VWA1	5.145 (4.880–5.544)	4.985 (4.497–5.479)	1.605 (0.748–3.445); 0.225
FLT3	2.666 (2.469–2.914)	2.890 (2.611–3.112)	0.198 (0.053–0.741); 0.016
COL9A1	1.159 (0.880–1.520)	1.595 (1.043–2.031)	0.398 (0.184–0.861); 0.019
BRK1	1.457 (1.167–1.692)	1.324 (1.118–1.458)	4.147 (1.076–15.988); 0.039
NELL1	8.846 (8.385–9.075)	8.883 (8.633–9.226)	0.492 (0.188–1.29); 0.149
SFTPA1	1.152 (0.889–1.426)	1.079 (0.835–1.282)	1.01 (0.539–1.895); 0.974
VPS37A	0.873 (0.615–1.420)	0.555 (0.395–0.919)	2.768 (1.249–6.135); 0.012
DRG2	1.151 (0.601–1.674)	0.721 (0.434–1.236)	1.878 (1.037–3.400); 0.037
HMBS	3.947 (3.284–4.464)	3.489 (2.393–4.040)	1.840 (1.121–3.017); 0.016
CLIP2	2.185 (1.866–2.682)	2.286 (1.661–2.622)	0.966 (0.584–1.597); 0.891
HSPB6	6.899 (6.393–7.351)	6.128 (5.733–6.771)	5.084 (2.308–11.198); 0.000
ATP6V1D	1.075 (0.881–1.581)	0.950 (0.771–1.149)	3.140 (1.198–8.230); 0.020
LACTB2	4.843 (4.193–5.294)	4.115 (3.522–4.703)	1.955 (1.205–3.171); 0.007
HBQ1	4.388 (3.830–4.951)	3.911 (2.598–5.099)	1.440 (1.029–2.016); 0.033
SCLY	6.124 (5.682–6.764)	5.778 (5.538–6.111)	3.588 (1.504–8.56); 0.004
MYO9B	1.528 (1.094–2.021)	1.251 (0.923–1.775)	1.401 (0.815–2.408); 0.223
CD300E	4.070 (3.832–4.544)	3.779 (3.583–4.164)	4.269 (1.611–11.311); 0.004
CDHR2	2.589 (1.804–3.275)	1.132 (0.693–2.067)	2.820 (1.712–4.644); 0.000
CPVL	4.479 (4.015–4.876)	4.811 (4.555–4.915)	0.295 (0.110–0.791); 0.015
ICAM4	5.253 (4.979–5.448)	5.229 (4.960–5.545)	1.229 (0.536–2.818); 0.626
PSMD9	3.598 (2.964–4.455)	3.048 (2.400–3.848)	1.693 (1.121–2.557); 0.012
VPS53	0.483 (0.167–1.176)	0.221 (−0.016–0.712)	1.542 (0.917–2.592); 0.102
CALCOCO1	6.179 (5.618–6.844)	5.753 (4.989–6.350)	1.285 (0.892–1.850); 0.178
UBAC1	3.990 (3.465–4.495)	3.645 (3.041–4.149)	2.162 (1.154–4.053); 0.016
PTPRM	4.227 (3.989–4.393)	4.210 (4.138–4.367)	0.562 (0.117–2.706); 0.473
GALNT7	4.344 (4.166–4.576)	4.505 (4.259–4.645)	0.497 (0.129–1.915); 0.310
FLT1	4.235 (4.075–4.415)	4.157 (4.064–4.266)	3.489 (0.56–21.742); 0.181
VAT1	4.014 (3.678–4.483)	3.801 (3.528–4.051)	3.569 (1.323–9.627); 0.012
L1CAM	6.812 (6.638–7.042)	6.801 (6.658–6.970)	3.636 (0.589–22.451); 0.165
GPA33	6.351 (5.596–7.801)	6.893 (5.512–8.086)	0.952 (0.710–1.276); 0.743
HLA-E	0.831 (0.757–0.955)	0.775 (0.611–0.863)	13.044 (1.56–109.096); 0.018
PCDH1	5.926 (5.777–6.047)	5.837 (5.742–5.953)	18.508 (1.399–244.885); 0.027
NPY	3.913 (3.519–4.458)	4.042 (3.690–4.556)	0.758 (0.416–1.381); 0.365

Note: IQR = Interquartile Range (25th and 75th centiles); OR = Odds Ratio; CI = Confidence Interval.

**Table 6 biomedicines-10-01857-t006:** Multivariate logistic regression model including all variables from the oncology panel with *p* < 0.02 at binary logistic regression.

Protein	OR (95% CI)	*p*-Value
CGB3	1.938 (0.586–6.415)	0.586
HSPB6	0.626 (0.062–6.360)	0.062
CDHR2	2.752 (0.937–8.085)	0.937
NCS1	41.559 (0.371–4649.940)	0.371
DCTPP1	1.713 (0.016–178.523)	0.016
LMLN	3.302 (0.987–11.044)	0.987
ALPP	0.739 (0.262–2.078)	0.262
SCLY	41.021 (0.694–2425.919)	0.694
CD300E	1.863 (0.125–27.774)	0.125
rfng	91.252 (0.288–28,947.58)	0.288
LACTB2	0.478 (0.022–10.402)	0.022
CCT5	0.003 (4.95 × 10^−6^–1.415)	0.000
GFER	0.156 (0.010–2.423)	0.010
VPS37A	0.352 (0.033–3.791)	0.033
VAT1	3.957 (0.053–296.511)	0.053
PSMD9	67.170 (1.007–4479.84)	1.007
KLK4	3.246 (0.433–24.334)	0.433
CPVL	0.103 (0.007–1.423)	0.007
FLT3	0.002 (9.21 × 10^−6^–0.398)	0.000
HMBS	0.500 (0.012–21.041)	0.012
UBAC1	0.998 (0.013–74.526)	0.013
HLA-E	0.252 (0.001–62.466)	0.001
AFP	0.771 (0.138–4.319)	0.138
COL9A1	0.054 (0.004–0.774)	0.004

Note: OR = Odds Ratio; CI = Confidence Interval.

**Table 7 biomedicines-10-01857-t007:** Oncology panel: result of the stepdown procedure applied to the saturated multivariate logistic regression model.

	*p*-Value	OR (95% CI)	Coefficient (95% CI)
CDHR2	0.002	2.368 (1.372–4.085)	0.8618858 (0.3165523–1.407219)
NCS1	0.001	30.429 (4.28–216.339)	3.415408 (1.45397–5.376846)
MLN	0.026	1.914 (1.082–3.384)	0.6490679 (0.0790204–1.219115)
FLT3	0.011	0.054 (0.006–0.507)	−2.915442 (−5.151296–−0.6795876)
COL9A1	0.034	0.285 (0.089–0.909)	−1.256405 (−2.417216–−0.0955934)
Constant	0.002	3.52 × 10^−10^ (4.36 × 10^−16^–2.84 × 10^−4^)	−21.76806 (−35.36822–−8.167891)

Note: OR = Odds Ratio; CI = Confidence Interval; Coefficient = logistic regression model coefficient.

**Table 8 biomedicines-10-01857-t008:** Oncology panel: sensitivity/specificity plot for the resulting multivariate logistic regression model, and for sensitivity levels higher than specificity levels.

Predicted Probability Cut Point	Sensitivity	Specificity
≥0.1731692	100.00%	55.81%
≥0.2520203	97.73%	62.79%
≥0.3550157	95.45%	69.77%
≥0.5128943	88.64%	76.74%
≥0.5453203	84.09%	79.07%

**Table 9 biomedicines-10-01857-t009:** Multivariate logistic regression model including all variables from the final model of the immuno-oncology and oncology panels.

Protein	OR (95% CI)	*p*-Value
Gal-1	99.147 × 10^4^ (65.740–1.50 × 10^10^)	0.005
MMP7	40.640 × 10^4^ (6.623–2.49 × 10^10^)	0.022
Gal-9	4.82 × 10^−4^ (8.37 × 10^−7^–0.277)	0.018
FASLG	1.665 × 10^−2^ (2.505 × 10^−4^–1.107)	0.056
CDHR2	1.738 (0.426–7.083)	0.441
NCS1	5.054 (0.252–101.451)	0.290
MLN	2.360 (0.771–7.227)	0.132
FLT3	6.315 × 10^−3^ (4.39 × 10^−5^–0.909)	0.046
COL9A1	3.174 × 10^−2^ (1.50 × 10^−3^–0.673)	0.027

Note: OR = Odds Ratio; CI = Confidence Interval.

**Table 10 biomedicines-10-01857-t010:** Joint panel: result of the stepdown procedure applied to the saturated multivariate logistic regression model.

	*p*-Value	OR (95% CI)	Coefficient (95% CI)
Gal-1	0.001	7.30 × 10^4^ (7.38 × 10^1^–7.21 × 10^7^)	11.198 (4.302–18.094)
MMP7	0.001	1.04 × 10^4^ (4.71 × 10^1^–2.29 × 10^6^)	9.249 (3.852–14.646)
Gal-9	0.010	8.98 × 10^−3^ (2.47 × 10^−4^–3.27 × 10^−1^)	−4.713 (−8.307–−1.119)
FASLG	0.003	1.56 × 10^−2^ (9.58 × 10^−4^–2.53 × 10^−1^)	−4.163 (−6.951–−1.375)
COL9A1	0.008	6.80 × 10^−2^ (9.34 × 10^−3^–4.96 × 10^−1^)	−2.688 (−4.673–−0.702)
Constant	0.001	1.72 × 10^−53^ (2.49 × 10^−83^–1.18 × 10^−23^)	−121.497 (−190.203–−52.791)

Note: OR = Odds Ratio; CI = Confidence Interval; Coefficient = logistic regression model coefficient.

**Table 11 biomedicines-10-01857-t011:** Joint panel: sensitivity/specificity plot for the resulting multivariate logistic regression model, and for sensitivity levels higher than specificity levels.

Predicted Probability Cut Point	Sensitivity	Specificity
≥0.2436392	100.00%	79.07%
≥0.3154508	97.67%	83.72%
≥0.4752594	95.35%	90.70%

## Data Availability

The data presented in this study are available on request from the corresponding author. The data are not publicly available due to ethical reasons.

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
