# Peer review of "A Targeted Proteomics Approach for Screening Serum Biomarkers Observed in the Early Stage of Type I Endometrial Cancer"

_biomedicines, 2022, doi:10.3390/biomedicines10081857_

Round 1

Reviewer 1 Report

This article is aimed to investigate proximity extension assay (PEA) for target biomarker analysis in serum of patients with endometrial cancer by using two commercialized oncology panels from OLINK company. Because the panels and the proximity extension assay (PEA) technology used in this study is a commercially available product, a potential bias cannot be totally ruled out. Although I appreciate the authors’ effort to generate possible proteomics to differentiate endometrial cancer from normal control and the data on this topic is always welcomed, however, these data need to be validated via basic laboratory work to draw reliable conclusions. This topic is interesting, and the following questions need clarification.

1.     Line 77: The authors did not explain why they chose patients with type I endometrial cancer as study population. Because type I endometrial cancer has relatively indolent behavior, whereas type II endometrial cancer has relatively aggressive behavior. That is, patients with type I endometrial cancer have more chance to be detected with precancerous lesion than type II endometrial cancer, contributing to the uncertainty of their role on this topic. By contrast, grade 3 endometrioid cell type may be an alternative choice based on a relatively poor prognosis, I recommend that the authors can consider to present the detail information of serum protein biomarkers of these patients.

2.     Line 86-88: There is no explanation why the author exclude patients with myoma and adenomyosis or HBV/HIV/HCV infection?

3.     Line 93-95: What is the difference between the immuno-oncology panel and the Target 96 oncology III panel? These panels focus on which types of malignancy?

4.     Line 98: What were the three negative controls used in this study? Only one was found.

5.     Line 133: Define “step-down procedure.”

6.     Line 158: Define “pseudo-R-squared.”

7.     Line 173-176 and statistical analyses: The authors excluded all proteins with more than 25% values below the limit of detection (LOD). The reason for the cut-off of 25% below the limit of detection should be explained.

8.     Line 293-296: The set of proteins Gal-1, Gal-9, MMP7, FASLG, and COL9A1 were identified and could separate endometrial cancer from control, and the AUC=96.9%. However, the sample size (total of 99 women, 44 from EC, and 44 non-EC control) might be inadequate to generalize the overall population. Additionally, is there any relationship in these biomarkers (Gal-1, Gal-9, MMP7, FASLG, COL9A1)? This should be stated in the limitations.

9.     The discussion of the identified proteins in this study is somewhat weak. The authors perform the Profiler tool and Reactome tool to categorize proteins into different pathways and molecular functions but result in highly unspecific results, therefore it is difficult to say that involvement of the proteins of both panels in the immune system in influencing tumor growth as described in lines 296-297.

10.  These are few descriptions about the association between these biomarkers and endometrial cancer.

11.  Overall, these is few clinical application value of this study because patients with endometrial cancer is usually diagnosed by the appearance of symptoms such as vaginal bleeding.

12.  Conclusion: I am not convinced that meaningful conclusions can be made from the data presented without validation. Why only the Immuno-oncology panel and Target 96 Oncology III panel were used among so many commercially available panels?

Author Response

Reviewer 1

Line 77: The authors did not explain why they chose patients with type I endometrial cancer as study population. Because type I endometrial cancer has relatively indolent behavior, whereas type II endometrial cancer has relatively aggressive behavior. That is, patients with type I endometrial cancer have more chance to be detected with precancerous lesion than type II endometrial cancer, contributing to the uncertainty of their role on this topic. By contrast, grade 3 endometrioid cell type may be an alternative choice based on a relatively poor prognosis, I recommend that the authors can consider to present the detail information of serum protein biomarkers of these patients.

Our reply: Endometrial cancer type I and II have no serum biomarker. Type I EC comprise the large majority of endometrial cancers, ~90%, while the type II EC comprise ~10%. In type I EC, stage 1 is the most frequent. For this reason, in this study we focused on type I EC at stage 1 patients to identify candidate serum protein biomarkers. The identification of biomarkers at an early stage would lead to a prompt diagnosis. A non-invasive test that can identify EC and reduce inappropriate and invasive examinations and improve patient care and prognosis. We agree with the Reviewer that it might be interesting to investigate grade 3 Type I ECs. Unfortunately, however, we do not have a sufficiently large number of cases to conduct the analyses on this group of patients.

Line 86-88: There is no explanation why the author exclude patients with myoma and adenomyosis or HBV/HIV/HCV infection?

Our reply:  We excluded controls patients with benign tumors (myoma), chronic inflammatory disease (adenomyosis) or viral infections because these pathologies may influence the abundance of several proteins in serum and, consequently, affect the proteomic analysis.We added these details in the methods section.

 Line 93-95: What is the difference between the immuno-oncology panel and the Target 96 oncology III panel? These panels focus on which types of malignancy?

Our reply: We thank the reviewer for this question. Olink Target 96 Immuno-Oncology panels include proteins associated with biological functions linked to immune response and immuno-oncology diseases. The biomarkers in this panel include proteins involved in processes such as promotion and inhibition of tumor immunity, chemotaxis, vascular & tissue remodeling, apoptosis & cell killing and metabolism & autophagy. Olink Target 96 oncology III panel comprises 92 human proteins that participate in biological mechanisms that are central to the initiation and progression of cancer, e.g., angiogenesis, cell communication, cellular metabolic pro-cesses, apoptosis, cell proliferation/differentiation etc. In Supplement Tables 2 and 3 all the proteins that make up the two panels are reported. These panels do not focus on specific malignancies. The categorization of the proteins included in the panel was carried out via referral to widely used public-access bioinformatic databases, including Uniprot, Human Protein Atlas, Gene Ontology (GO) and DisGeNET. These details are now reported in the methods section.

Line 98: What were the three negative controls used in this study? Only one was found.

Our reply: We added more details in the methods about this aspect.

Line 133: Define “step-down procedure.”

Our reply: A step-down procedure starts from a saturated model – which is a model including all studied independent variables – and pulls out of the model, one at the time, the variables with the highest p-value. The procedure end when all the independent variables left in the model are significantly associated with the dependent variable. How the procedure is conducted is explained in the Methods section.

Line 158: Define “pseudo-R-squared.”

Our reply: A Pseudo-R-Squared is a measure of goodness-of-fit used in logistic regression, where the equivalent R-Squared used in ordinary least square (OLS) regression cannot be used. The measure varies between 0 and 1, with higher values indicating better fit.

Line 173-176 and statistical analyses: The authors excluded all proteins with more than 25% values below the limit of detection (LOD). The reason for the cut-off of 25% below the limit of detection should be explained.

Our reply: Several of the investigated proteins had many values below the limit of detection (LOD). We choose to exclude from the proteins that had more than 25% values under the LOD. Olink suggests to exclude assays in the range of less than 25-50% of samples above LOD (https://www.olink.com/faq/how-is-the-limit-of-detection-lod-estimated-and-handled/), but we adopted a more restrictive approach and chose to have at least 75% of samples above the LOD.

Line 293-296: The set of proteins Gal-1, Gal-9, MMP7, FASLG, and COL9A1 were identified and could separate endometrial cancer from control, and the AUC=96.9%. However, the sample size (total of 99 women, 44 from EC, and 44 non-EC control) might be inadequate to generalize the overall population. Additionally, is there any relationship in these biomarkers (Gal-1, Gal-9, MMP7, FASLG, COL9A1)? This should be stated in the limitations.

Our reply: The number of patients used in this study is not a large population. But still, with 88 women 44 from EC, and 44 non-EC control we obtained a good results AUC=96.9%. A larger sample would have certainly allowed to include more proteins in the saturated model, but still the result in very satisfactory. If the identified proteins were related to one-another, in terms of variability, they would have not been included in the same model, due to collinearity. The fact that these proteins stay together in the model, implies they explain variability in an independent way. We added: Another limit of our study is the small number of patients which does not allow us to generalize to the overall population. Even if the results are satisfactory, more studies are certainly required to confirm and consolidate these findings.

The discussion of the identified proteins in this study is somewhat weak. The authors perform the Profiler tool and Reactome tool to categorize proteins into different pathways and molecular functions but result in highly unspecific results, therefore it is difficult to say that involvement of the proteins of both panels in the immune system in influencing tumor growth as described in lines 296-297.

Our reply: According to the indications of the reviewer, lines 296-297 have been removed.

These are few descriptions about the association between these biomarkers and endometrial cancer.

Our reply: As this reviewer correctly points out, for many of the identified biomarker an involvement in EC have been reported. We have extended the discussion, adding this description.

Overall, these is few clinical application value of this study because patients with endometrial cancer is usually diagnosed by the appearance of symptoms such as vaginal bleeding.

Our reply: We thank the reviewer for the questions. We want to clarify that bleeding alone is not enough to diagnose an EC. There is a well-defined procedure that begins with the assessment of the symptoms, and proceeds with pelvic or transvaginal ultrasound (TVUS) and finally with hysteroscopy. Endometrial tissue can be sampled by targeted endometrial biopsy with hysteroscopy. By using serum biomarkers, we can lead to a faster diagnosis and reduce inappropriate and invasive examinations.

Conclusion: I am not convinced that meaningful conclusions can be made from the data presented without validation. Why only the Immuno-oncology panel and Target 96 Oncology III panel were used among so many commercially available panels?

Our reply: We agree with the reviewer regarding the comment on data validation. Data validation will be done in future studies. This is a first promising step. We identified biomarkers that, together, have a very strong predictive value. Of course, these results need to be further validated. We used these panels for the proteins that they contain and because they are the latest developed by Olink.

Reviewer 2 Report

The authors searched for serum biomarker proteins for diagnosing the early stages of endometrial cancer (EC) using the proximity extension assay (PEA) of the Immuno-oncology panel and the Target 96 Oncology III panel provided by Olink Proteomics. They found 20 and 24 proteins associated with EC from the former and latter panels. By combining the candidate proteins from two panels with a multivariate logistic regression model, they generated a model with a sensitivity of 97.67% and a specificity of 83.72%. They concluded that they succeeded in developing an algorism discriminating early EC patients from normal subjects using a small amount of blood sample. Further studies will be required to validate the results with age-matched cases and controls to establish a diagnostic technique of EC using a small serum sample.

The results seem to be reasonable. However, the authors should solve the following issues before considering the publication of the manuscript in Biomedicines.

Introduction

1.       Why did the authors try to find serum bio-markers specific to the early stage of EC? Is screening or diagnosing EC essential for supporting women’s health?

2.       Why did they focus on type-I EC? First, it is associated with a good prognosis (line 45). Second, all EC patients used in the study were diagnosed with type-I EC (line 77 in the Materials and Methods section). Isn’t it beneficial to find serum biomarkers to diagnose type II EC, which is a poor prognosis?

3.       Why did they apply proximity extension assays (PEA)? Please describe the merits of PEA in more detail with references.

Materials and Methods

1.       Line 90, the title should be “Serum sample collection and the PEA.”

2.       Please describe the manufacturer information and characteristics of Immuno-oncology and target 96 oncology III panels.

3.       Line 110, what is zanalysed?

4.       Line 111, what is zcategorized?

5.       Line 110, please describe g:Profiler in more detail with its URL and references.

6.       Line112, please describe Reactome in more detail with its URL and references.

Results

1.       Is the Bioinformatic analysis (Lines 244-276) sub-section necessary to lead the authors’ conclusion of the manuscript? The analysis may help find the mechanism of developing EC; however, the study aimed to find the EC-associated serum biomarkers. Do they want to find other biomarkers using the analysis?

2.       Lines 251 and 255, what is zcategorized?

Discussion

The description of the discussion section seems to be redundant. Repetition of the results also exists. Please revise the discussion section with compacting quotation parts and reinforcing their results and merit using PEA with the previous reports finding biomarkers using PEA. Furthermore, the reviewer wonders whether the biomarkers found in this study are truly EC-specific. These biomarkers might be rather a cancer/tumor-specific, not EC-specific.

Conclusion

This section should not be contained the authors’ wishful outlook. A sentence in lines 330-332 should be removed.

Author Response

Reviewer 2

Why did the authors try to find serum bio-markers specific to the early stage of EC? Is screening or diagnosing EC essential for supporting women’s health?

Our reply: The early stage of EC is the most frequent: ~80%. Diagnosis at an early stage is essential. A non-invasive test that can identify EC and reduce inappropriate invasive examinations could transform patient care. In addition, early diagnosis improves patients' chances of survival, which are around 90% at five years in the first stage.

Why did they focus on type-I EC? First, it is associated with a good prognosis (line 45). Second, all EC patients used in the study were diagnosed with type-I EC (line 77 in the Materials and Methods section). Isn’t it beneficial to find serum biomarkers to diagnose type II EC, which is a poor prognosis?

Our reply: To date, type-I EC does not have identified serum biomarkers. Type-I ECs comprise the large majority of endometrial cancers: ~90%. 80% of patients are in the early stages while only 20% are in more advanced phases. Patients in more advanced stages require more specific care, such as radio therapy or chemotherapy. For this purpose, identification of biomarkers at an early stage is important. Many patients discovered in an early stage will not need to be subjected to a harsh treatment. We agree with the reviewer on the comment about biomarkers’ discovery for type-II EC, but at present we do not have enough patients to conduct such a study.

Why did they apply proximity extension assays (PEA)? Please describe the merits of PEA in more detail with references.

Our reply: We choose Olink proteomics panels because of the biological processes of proteins that are involved. In the third question of the first reviewer, we list the types of biological processes in which these proteins are involved. The Olink PEA technology is an innovative dual recognition, DNA-coupled methodology, providing exceptional readout specificity. PEA enables high multiplex, rapid throughput biomarker analysis without compromising on data quality. Homogeneous antibody-based proximity extension assays provide sensitive and specific detection of low-abundant proteins in the human blood. The merits of PEA are explained in more detail in the introduction.

Materials and Methods

 Line 90, the title should be “Serum sample collection and the PEA.”

Our reply: We corrected the title.

Please describe the manufacturer information and characteristics of Immuno-oncology and target 96 oncology III panels.

Our reply: The manufacturer information and characteristics of Immuno-oncology and target 96 oncology III panels have been added in the materials and methods.

Line 110, what is zanalysed?

Our reply: It was a mistake we have now fixed.

Line 111, what is zcategorized?

Our reply: It was a mistake we have now fixed.

Line 110, please describe g:Profiler in more detail with its URL and references.

Our reply: We added an URL and the references.

Line112, please describe Reactome in more detail with its URL and references.

Our reply: We added an URL and the references.

Results

Is the Bioinformatic analysis (Lines 244-276) sub-section necessary to lead the authors’ conclusion of the manuscript? The analysis may help find the mechanism of developing EC; however, the study aimed to find the EC-associated serum biomarkers. Do they want to find other biomarkers using the analysis?

Our reply: The Bioinformatic analysis is necessary for us to make an enrichment data analysis. This analysis helps us identify the likely mechanisms in which these proteins participate in the development of the EC. We did not use the bioinformatic analysis to draw conclusions. We do not want to find other biomarkers using the bioinformatic analysis.

Lines 251 and 255, what is zcategorized?

Our reply: It was a mistake we have now fixed.

Discussion

The description of the discussion section seems to be redundant. Repetition of the results also exists. Please revise the discussion section with compacting quotation parts and reinforcing their results and merit using PEA with the previous reports finding biomarkers using PEA. Furthermore, the reviewer wonders whether the biomarkers found in this study are truly EC-specific. These biomarkers might be rather a cancer/tumor-specific, not EC-specific.

Our reply: In this study we used EC patients without other oncological pathologies. Unfortunately, there are no specific panels for EC. Obviously other studies are needed to better understand if these proteins are directly related to the EC. We have broadened the discussion.

Conclusion

This section should not be contained the authors’ wishful outlook. A sentence in lines 330-332 should be removed.

Our reply: The sentence in lines 330-332 was removed.

Reviewer 3 Report

To authors,

1.     Abstract. Please state that all patients were of type I EC.

2.     Ages are different between the two groups. Usually, we choose control women with “same/similar age” group (sometimes 1:1 allocation; because there are abundant candidate control women). Please state why you did not do this and that this (not doing it) does not much distort/bias the data/result/interpretation. You touched this at the limitation paragraph; however, to me, to pick up similar aged control women (1:1 for example) may not so difficult. Please explain why it SO difficult. 

After surgery, these protein levels decrease? Let us assume that some decease and others not. Then, the former might be secreted from the EC itself, whereas the latter may indicate the “tendency” “background” of the women who tend to suffer from EC. I do not know if this is a good example; let us assume that some adipokine increases in EC patient serum; this can be either due to EC itself OR abundance of adipose tissue (obesity); with the latter is a strong risk/background of EC-occurrence. The similar might occur in your data. Please touch this possibility if necessary.  

Author Response

Reviwer 3

Abstract. Please state that all patients were of type I EC.

Our reply: We added that all patients are type I EC.

Ages are different between the two groups. Usually, we choose control women with “same/similar age” group (sometimes 1:1 allocation; because there are abundant candidate control women). Please state why you did not do this and that this (not doing it) does not much distort/bias the data/result/interpretation. You touched this at the limitation paragraph; however, to me, to pick up similar aged control women (1:1 for example) may not so difficult. Please explain why it SO difficult.

Our reply: We thank the reviewer for the questions. We have not been able to find women with similar ages to patients with EC because it is difficult to find “healthy” women of that age in a hospital. Women with an average age of 67, that refer to hospital, are very difficult to be found healthy. Moreover, women at that age usually have different kinds of comorbidities which can affect the abundance of protein in the serum. On the contrary, young women that refer to hospital, usually are healthy persons and come to hospital also for check-up, for example infertility or dysfunctional hormonal problem. Finally, endometrial biopsies of fertile patients are more abundant respect thin endometrial layer in menopausal patients (as in patients with median age of 67 years old).

After surgery, these protein levels decrease? Let us assume that some decease and others not. Then, the former might be secreted from the EC itself, whereas the latter may indicate the “tendency” “background” of the women who tend to suffer from EC. I do not know if this is a good example; let us assume that some adipokine increases in EC patient serum; this can be either due to EC itself OR abundance of adipose tissue (obesity); with the latter is a strong risk/background of EC-occurrence. The similar might occur in your data. Please touch this possibility if necessary.  

Our reply: We thank the reviewer for this comment. Main aim of this work is to find early EC diagnostic biomarker, for this reason all the blood samples were taken before surgery.  At this stage we cannot exclude that surgery could affect the abundance of the tested proteins. Unfortunately, we did not measure proteins after surgery; it should be of interest to identify marker to be used not only for diagnosis but also for follow-up; to date this goes far beyond the scope of this work, but it can be exploited in future studies.

Round 2

Reviewer 1 Report

Since the topic is related to stage I and type I endometrial cancer, please use the “precise” title for this. For example, endometrioid type endometrial cancer may be better than endometrial cancer.

Since grade and differentiation as well as surgical pathological parameters, such as more than half of myometrial invasion, presence of lymphovascular space involvement, and others, may be one of the critical points for the prognosis of the early stage endometrioid endometrial cancer, and additionally, the authors would like to present the value about the prognosis, please describe it or explain it.

English typing errors can be improved. 

some unclear parts need clarification, and strength and limitations should be mentioned clearly. 

Author Response

Reviwer 1

Since grade and differentiation as well as surgical pathological parameters, such as more than half of myometrial invasion, presence of lymphovascular space involvement, and others, may be one of the critical points for the prognosis of the early stage endometrioid endometrial cancer, and additionally, the authors would like to present the value about the prognosis, please describe it or explain it.

Our reply:

The 2020 ESGO/ESTRO/ESP guidelines on EC define 5 prognostic groups based on stage, degree of differentiation, histotype, and invasion of lymph vascular spaces. The Groups are classified as low, intermediate, intermediate-high, high and metastatic. Stage I Endometrioid endometrial carcinoma falls into the first three groups depending on the sub-stage and degree of differentiation. In particular, G1 and G2 are considered low grades, while G3 high grade.

A better classification can be done if the molecular characteristics of the lesion are taken into account; this is currently done by  immunohistochemical  analysis of three markers (p53, MSH6 and PMS2) and by mutation analysis of the exonuclease domain of POLE. However, the guidelines do not provide precise OS and PFS values for different groups. In the supplementary table we reported the corresponding risk groups.

Reviewer 2 Report

The authors responded to the reviewer’s query in their reply comments; however, the authors did not reflect some of them in the revised manuscript. In addition, their results did not identify type I EC-specific serum markers. Therefore, the reviewer feels uncomfortable with the title of the manuscript. “A targeted proteomics approach for screening serum biomarkers observed in the early stage of endometrial cancer” can be a suitable title.

Introduction

Some serum markers like CEA, CA19-9, and CA125 for screening EC already exist, although these serum markers are not specific to EC. If the authors want to identify new EC-specific serum markers, please describe the background, the necessity of serum markers for diagnosing type I EC, and the benefit of identifying the type I EC-specific serum marker in detail.

Results

The authors replied that the bioinformatic analyses were necessary for enrichment data analysis. However, these data were not reflected in the authors’ conclusions at all. If the authors assert their necessity, please add descriptions of how they avail these data to enrich their results and conclusion in the discussion section. Available bioinformatic data seem specific to cancer progression, metastasis, and inflammation, not to the early stages of EC.

Discussion

The five-hit proteins, Gal-1, Gal-9, MMP7, COL9A1, and FASLG, are often overexpressed in several tumors. Therefore, in the reviewer’s opinion, they found five proteins involved in oncogenesis and cancer immunoediting in the serum from type I EC using the PEA. Although the authors added the limitation of their study in lines 397-399 in the discussion section, do they really think the above five proteins can be serum markers for the early stage of EC? Please describe the possibility and the feasibility of their findings concerning the development of type I EC-specific serum markers with the study’s limitation at length.

Author Response

Reviwer 2

The authors responded to the reviewer’s query in their reply comments; however, the authors did not reflect some of them in the revised manuscript. In addition, their results did not identify type I EC-specific serum markers. Therefore, the reviewer feels uncomfortable with the title of the manuscript. “A targeted proteomics approach for screening serum biomarkers observed in the early stage of  endometrial cancer” can be a suitable title.

Our reply: We thanks the reviewer for the title improvement. We substituted the title with: “A targeted proteomics approach for screening serum biomarkers observed in the early stage of type I endometrial cancer”.

Introduction

Some serum markers like CEA, CA19-9, and CA125 for screening EC already exist, although these serum markers are not specific to EC. If the authors want to identify new EC-specific serum markers, please describe the background, the necessity of serum markers for diagnosing type I EC, and the benefit of identifying the type I EC-specific serum marker in detail.

Our reply: We added more details in the background  about this aspect.

Results

The authors replied that the bioinformatic analyses were necessary for enrichment data analysis. However, these data were not reflected in the authors’ conclusions at all. If the authors assert their necessity, please add descriptions of how they avail these data to enrich their results and conclusion in the discussion section. Available bioinformatic data seem specific to cancer progression, metastasis, and inflammation, not to the early stages of EC.

Our reply: We added more details in the discussion  about this aspect.

Discussion

The five-hit proteins, Gal-1, Gal-9, MMP7, COL9A1, and FASLG, are often overexpressed in several tumors. Therefore, in the reviewer’s opinion, they found five proteins involved in oncogenesis and cancer immunoediting in the serum from type I EC using the PEA. Although the authors added the limitation of their study in lines 397-399 in the discussion section, do they really think the above five proteins can be serum markers for the early stage of EC? Please describe the possibility and the feasibility of their findings concerning the development of type I EC-specific serum markers with the study’s limitation at length.

Our reply:

As the reviewer point out, we are aware that Gal-1, Gal-9, MMP7, COL9A1, and FASLG are broadly overexpressed in different kind of cancer, and indeed for this study we used the Immuno-oncology panel and the Target 96 Oncology III panel including 92 proteins already associated with tumorigenesis. Exploiting these panels in 44 healthy controls and 44 type I EC at tumor stage 1 among the 92 cancer-related proteins we found that levels of Gal-1, Gal-9, MMP7, COL9A1, and FASLG combined together can discriminate between healthy control and woman affected by type I EC. We are convinced that analysis of serum level of the five proteins, combined with a clinical manifestation typical of EC, such us bleeding and pelvic pain, might help early EC diagnosis avoiding invasive diagnostic techniques.  These consideration were added in the conclusion section of the manuscript.

Round 3

Reviewer 2 Report

In the second revised manuscript version, the authors responded to the reviewer’s queries and reflected in the revision.

The reviewer requests one more thing to revise for considering publication.

Lines 95-96, this sentence can be replaced with “Our study aims to apply the PEA technology to screen candidate serum protein markers of early-stage of type I EC.”

Author Response

Lines 95-96, this sentence can be replaced with “Our study aims to apply the PEA technology to screen candidate serum protein markers of early-stage of type I EC.”

Our reply: According to the indications of the reviewer lines 95-96 were changed